# A SoxB gene acts as an anterior gap gene and regulates posterior segment addition in a spider

Christian Louis Bonatto Paese[1], Anna Schoenauer[1], Daniel J Leite[1], Steven Russell[2], Alistair P McGregor[1]*

[1]Laboratory of Evolutionary Developmental Biology, Department of Biological and Medical Sciences, Oxford Brookes University, Oxford, United Kingdom; [2]Department of Genetics, University of Cambridge, Cambridge, United Kingdom

**Abstract** Sox genes encode a set of highly conserved transcription factors that regulate many developmental processes. In insects, the SoxB gene *Dichaete* is the only Sox gene known to be involved in segmentation. To determine if similar mechanisms are used in other arthropods, we investigated the role of Sox genes during segmentation in the spider *Parasteatoda tepidariorum*. While *Dichaete* does not appear to be involved in spider segmentation, we found that the closely related *Sox21b-1* gene acts as a gap gene during formation of anterior segments and is also part of the segmentation clock for development of the segment addition zone and sequential addition of opisthosomal segments. Thus, we have found that two different mechanisms of segmentation in a non-mandibulate arthropod are regulated by a SoxB gene. Our work provides new insights into the function of an important and conserved gene family, and the evolution of the regulation of segmentation in arthropods.

DOI: https://doi.org/10.7554/eLife.37567.001

*For correspondence:
amcgregor@brookes.ac.uk

**Competing interests:** The authors declare that no competing interests exist.

## Introduction

Arthropods are the most speciose and widespread of the animal phyla, and it is thought that their diversification and success are at least in part explained by their segmented body plan (*Tautz, 2004*). In terms of development, insects utilise either derived long germ embryogenesis, where all body segments are made more or less simultaneously, or short/intermediate germ embryogenesis, where a few anterior segments are specified and posterior segments are added sequentially from a growth or segment addition zone (SAZ) (*Peel et al., 2005*; *Davis and Patel, 2002*). It is thought that segmentation in the ancestral arthropod resembled the short germ mode seen in most insects (*Peel et al., 2005*; *McGregor et al., 2009*). Understanding the regulation of segmentation more widely across the arthropods is important for understanding both the development and evolution of these highly successful animals.

We have a detailed and growing understanding of the regulation of segmentation in various insects, especially the long germ dipteran *Drosophila melanogaster* and the short germ beetle *Tribolium castaneum.* However, studies of other arthropods including the myriapods *Strigamia maritima* and *Glomeris marginata*, and chelicerates, such as the spiders *Cupiennius salei* and *Parasteatoda tepidariorum*, have provided important mechanistic and evolutionary insights into arthropod segmentation (*Peel et al., 2005*; *Hilbrant et al., 2012*; *McGregor et al., 2008a*; *Leite and McGregor, 2016*; *Janssen et al., 2004*; *Brena and Akam, 2012*). Previous studies have shown that different genetic mechanisms are used to generate segments along the anterior-posterior axis of spider embryos. In the anterior tagma, the prosoma or cephalothorax, the cheliceral and pedipalpal segments are generated by dynamic waves of *hedgehog* (*hh*) and *orthodenticle* (*otd*) expression

**eLife digest** Insects, spiders, centipedes and lobsters all belong to a group of animals known as arthropods. A common feature of these animals is that their bodies are made up of repeated segments. However different arthropods build their segmented bodies in different ways. For example, the fruit fly makes all of its segments at the same time, while most other arthropods – including spiders – make a few segments at once and then add the rest, one or two at a time, to the rear end of their bodies.

Recent research in different insects has shown that these two processes – adding segments simultaneously or sequentially – are more similar than previously thought. This research also showed that these processes involve a gene called *Dichaete*, which belongs to the Sox gene family. However it was not known if Sox genes also control the production of segments in other arthropods like spiders.

Paese et al. have now found that, just like insects, the common house spider does indeed require a Sox gene to form its segments. Specifically, the experiments revealed that spiders need a Sox gene called *Sox21b-1* to make both the segments that carry their legs (which are made all at once), and the segments that make up the rear of their bodies (which are added one at a time).

Since spiders and insects both use a Sox gene to control the formation of their body segments, it is likely that the ancestor of arthropods used one too. However, because spiders and insects use a different Sox gene for these processes, Paese et al. suggest that one gene may have replaced the role of the other during the evolution of insects and spiders.

Together these findings broaden the current understanding of how genes interact to organise cells to build organisms and how these processes evolve over time. Furthermore, since Sox genes direct many important events in all animals, including humans, the discovery of a new role for one of these genes may help scientists to better understand the development of other animals too.

DOI: https://doi.org/10.7554/eLife.37567.002

(*Kanayama et al., 2011*; *Pechmann et al., 2009*). The leg-bearing segments are specified by gap gene like functions of *hunchback* (*hb*) and *distal-less* (*dll*) (*Pechmann et al., 2011*; *Schwager et al., 2009*). In contrast, the segments of the posterior tagma, the opisthosoma or abdomen, are generated sequentially from a SAZ. This process is regulated by dynamic interactions between Delta-Notch and Wnt8 signalling to regulate *caudal* (*cad*), which in turn is required for oscillatory expression of pair-rule gene orthologues including *even-skipped* (*eve*), and *runt* (*run*) (*McGregor et al., 2009*; *McGregor et al., 2008b*; *Schönauer et al., 2016*). Interestingly, these pair-rule gene orthologues are not involved in the production of the prosomal segments (*Schönauer et al., 2016*). Therefore, the genetic regulation of segmentation along the anterior-posterior axis in the spider exhibits similarities and differences to segmentation in both long germ and short germ insects.

The Group B Sox (Sry-Related High-Mobility Group box) family gene *Dichaete* is required for correct embryonic segmentation in the long germ insect *D. melanogaster*, where it regulates pair-rule gene expression (*Nambu and Nambu, 1996*; *Russell et al., 1996*). Interestingly, it was recently discovered that a *Dichaete* orthologue is also likely involved in segmentation in the short germ insect *T. castaneum* (*Clark and Peel, 2018*). This similarity is consistent with work inferring that these modes of segmentation are more similar than previously thought and provides insights into how the long germ mode evolved (*Clark and Peel, 2018*; *Clark, 2017*; *Verd et al., 2018*). However, it appears that despite these similarities, *Dichaete* can play different roles in *D. melanogaster* and *T. castaneum* consistent with the generation of segments simultaneously via a gap gene mechanism in the former and sequentially from a posterior SAZ in the latter (*Clark and Peel, 2018*).

We recently described the characterisation of 14 Sox genes in the genome of the spider *P. tepidariorum* (*Paese et al., 2017*) and that several of the spider Sox genes are represented by multiple copies likely produced during the whole genome duplication (WGD) in the lineage leading to this arachnid (*Paese et al., 2017*; *Schwager et al., 2017*). Interestingly, while *Dichaete* is not expressed in a pattern consistent with a role in segmentation (*Paese et al., 2017*), we found that the closely related SoxB gene, *Sox21b-1,* is expressed in both the prosoma and opisthosoma before and during segmentation (*Paese et al., 2017*). Here, we report that in *P. tepidariorum*, *Sox21b-1* regulates both

prosomal and opisthosomal segmentation. In the prosoma, *Sox21b-1* has a gap-like gene function and is required for the generation of the four leg-bearing segments. In addition, *Sox21b-1* appears to act upstream of both Delta-Notch and Wnt8 signalling to regulate the formation of the SAZ, and knockdown of *Sox21b-1* results in truncated embryos missing all opisthosomal segments. Therefore, while prosomal and opisthosomal segments are generated by different mechanisms in the spider, our analysis shows that *Sox21b-1* is required for segmentation in both regions of the developing spider embryo.

## Results

### *Sox21b-1* is maternally deposited and is subsequently expressed in the germ disc and germ band of spider embryos

We previously identified and assayed the expression of the complement of Sox genes in the genome of the spider *P. tepidariorum* (*Paese et al., 2017*) (and see *Figure 1*). Our phylogenetic analysis indicates that *P. tepidariorum Sox21b-1* and its paralog *Sox21b-2* are members of the Sox group B, closely related to the *Drosophila Dichaete* and *Sox21b* genes (*Figure 1—figure supplement 1*). In insects (*McKimmie et al., 2005*; *Wilson and Dearden, 2008*), *Dichaete, Sox21a* and *Sox21b* are clustered in the genome, however, both *Sox21b* paralogs are dispersed in the spider genome (*Paese et al., 2017*). This suggests that *Sox21b-1* and *Sox21b-2* possibly arose from the WGD event in the ancestor of arachnopulmonates (*Schwager et al., 2017*) rather than by a more recent tandem duplication (*Figure 1—figure supplement 1*).

In light of its interesting expression pattern, we elected to further analyse *Sox21b-1*. Pre-vitellogenic *P. tepidariorum* oocytes contain a Balbiani's body (*Jedrzejowska and Kubrakiewicz, 2007*), where maternally deposited factors are enclosed, and we found that *Sox21b-1* is abundant in this region, indicating that it is maternally contributed (*Figure 1A*). However, after fertilization we did not detect *Sox21b-1* mRNA until early stage 5, when weak expression is detected throughout the germ disc, with stronger expression in more central cells (*Figure 1B–C*). At late stage 5, expression becomes more restricted to the centre of the germ disc (*Figure 1D*). During stages 5 and 6, the cumulus migrates to the rim of the germ disc, opening the dorsal field and giving rise to an axially symmetric germ band (*Figure 1E*) (see *Mittmann and Wolff, 2012*). In early stage 6 embryos, *Sox21b-1* is observed in the middle of the presumptive prosoma in a broad stripe (*Figure 1E*), which develops further during stage 7 in the region where the leg-bearing segments will form (*Figure 1F*). This expression pattern resembles the previously described expression of the gap gene *hb* (*Schwager et al., 2009*).

During these and subsequent stages, dynamic expression of *Sox21b-1* is observed in the SAZ and the most anterior region of the germ band that will give rise to the head segments (*Figure 1H–I*). Later in development, the expression of *Sox21b-1* resembles that of *SoxNeuro* (*SoxN*), another Group B Sox gene (*Paese et al., 2017*). This expression is similar to that of both *SoxN* and *Dichaete* in the *D. melanogaster* neuroectoderm and segregating neuroblasts, however in spiders the neuroectoderm does not produce stem cell like neuroblasts, but instead clusters of delaminating cells that adopt the neural fate (*Paese et al., 2017*; *Stollewerk and Chipman, 2006*; *McKimmie et al., 2005*) (*Figure 1H–I*). Expression of the related group B Sox genes, *Dichaete* and *Sox21b-2* are not detected in *P. tepidariorum* during embryonic development (*Paese et al., 2017*). The expression of *Sox21b-1* in the embryo suggests that it is involved in both anterior and posterior segmentation in this spider, and then later during nervous system development.

### *Sox21b-1* regulates prosomal and opisthosomal segmentation

To assay the function of *Sox21b-1* during embryogenesis we knocked down the expression of the gene using a parental RNAi approach (*Akiyama-Oda and Oda, 2006*). We observed three phenotypic classes, which were consistent between both non-overlapping *Sox21b-1* fragments we used for RNAi (*Figure 2*, *Figure 2—figure supplement 1*, *Supplementaryfile 1*).

Class I embryos developed a presumptive head region (*Figure 2A–C*), as well as normal cheliceral, pedipalpal and first leg-bearing (L1) segments (*Figure 2C*). The identity of these segments was confirmed by expression of *labial* (*lab*) in the pedipalps and L1, and *Deformed-A* (*Dfd*-A) in L1 (*Figure 2—figure supplement 2A–D*). However, the other three leg-bearing segments, L2 - L4, as well

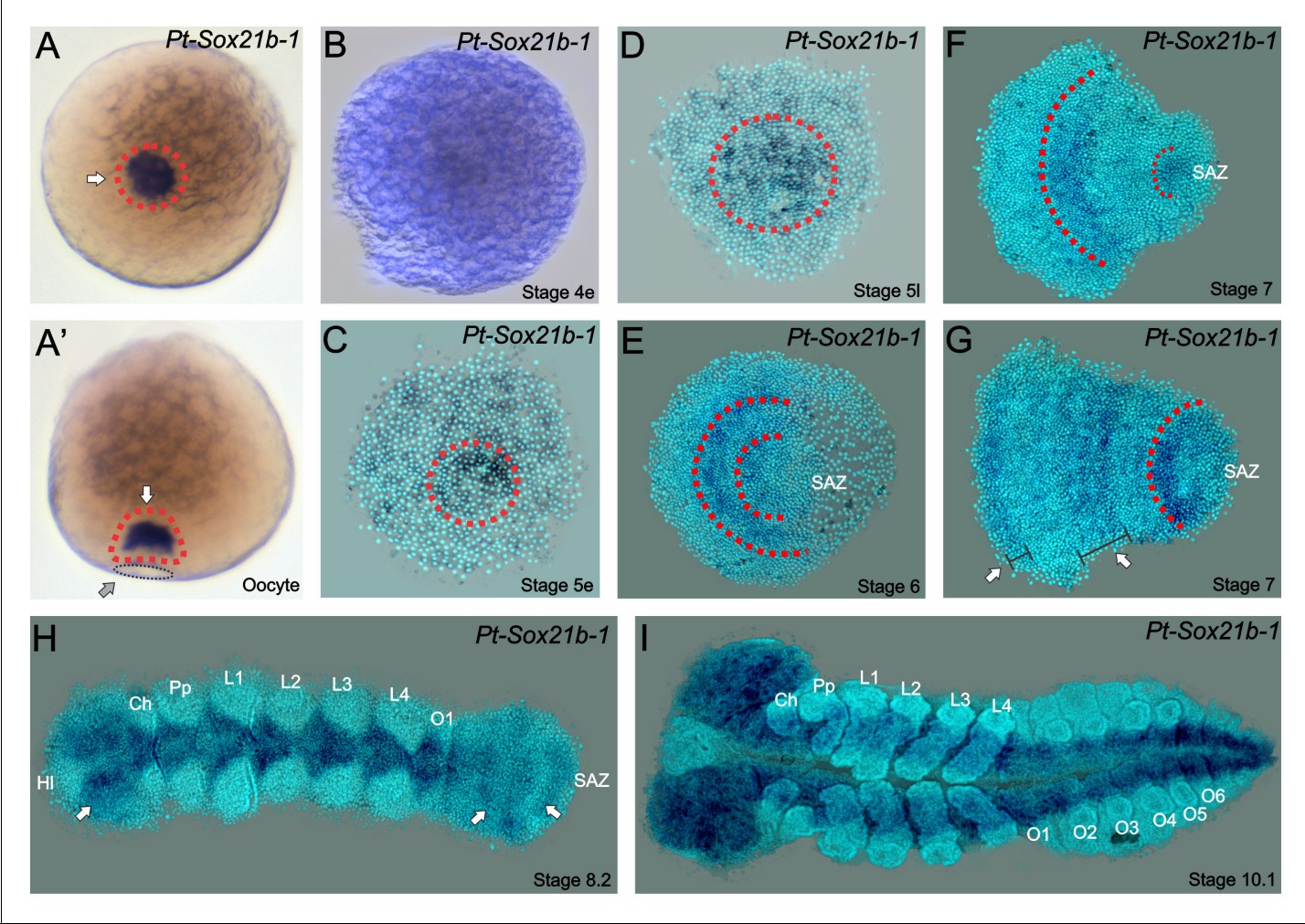

**Figure 1.** Expression of *Sox21b-1* in *P. tepidariorum* oocytes and embryos. (**A**) Dorsal (left) and lateral (right) views of pre-vitellogenic oocytes showing *Sox21b-1* mRNA in the Balbiani's body (red dashed circle and white arrows). The sperm implantation groove is indicated by a black dashed circle and grey arrow. (**B**) Overstained early stage 4 embryo evidencing the lack of expression of *Sox21b-1* at this stage. (**C**) At early stage 5, the expression of *Sox21b-1* appears in a salt and pepper pattern in the germ disc. (**D**) Expression in the cumulus becomes stronger at late stage 5, with less expression at the periphery of the germ disc (dashed red circle). (**E**) At stage 6 *Sox21b-1* is expressed in a broad stripe in the anterior (between the red dashed lines). (**F**) At stage 7 there is expression in the region of the presumptive leg-bearing segments and in the SAZ (both indicated by red dashed lines). (**G**) At late stage 7 *Sox21b-1* is still expressed in the SAZ (red dashed line) and the presumptive leg-bearing segments (indicated by a white arrow and wide black bracket), but nascent expression is observed at the anterior of the germ band (indicated by a white arrow and narrow black bracket). (**H**) At stage 8.2, when the limb buds are visible the expression of *Sox21b-1* becomes restricted to the ventral nerve cord (anterior white arrow) and can be observed in the SAZ (posterior arrows). (**I**) At stage 10.1, *Sox21b-1* expression is restricted to the ventral nerve cord and the head lobes. Ch: Chelicerae; Hl: Head lobes; L1 to L4: Prosomal leg- bearing segments 1 to 4; O1 to O6: Opisthosomal segments 1 to 6; SAZ: Segment Addition Zone. Ventral views are shown with anterior to the left, except as described for oocytes.

DOI: https://doi.org/10.7554/eLife.37567.003

The following source data and figure supplement are available for figure 1:

**Source data 1.** ClustalW alignment of Sox protein HMG domains.
DOI: https://doi.org/10.7554/eLife.37567.005
**Figure supplement 1.** RAxML phylogeny of eumetazoan Sox genes.
DOI: https://doi.org/10.7554/eLife.37567.004

as all of the opisthosomal segments were missing in Class I embryos. These embryos exhibited a truncated germ band, terminating in disorganised tissue in the region of the SAZ (*Figure 2C*). In the case of Class II phenotypes, embryos only differentiated the head region and the cheliceral and ped-ipalpal segments (*Figure 2D*, *Figure 2—figure supplement 2A–B*): all leg-bearing segments of the

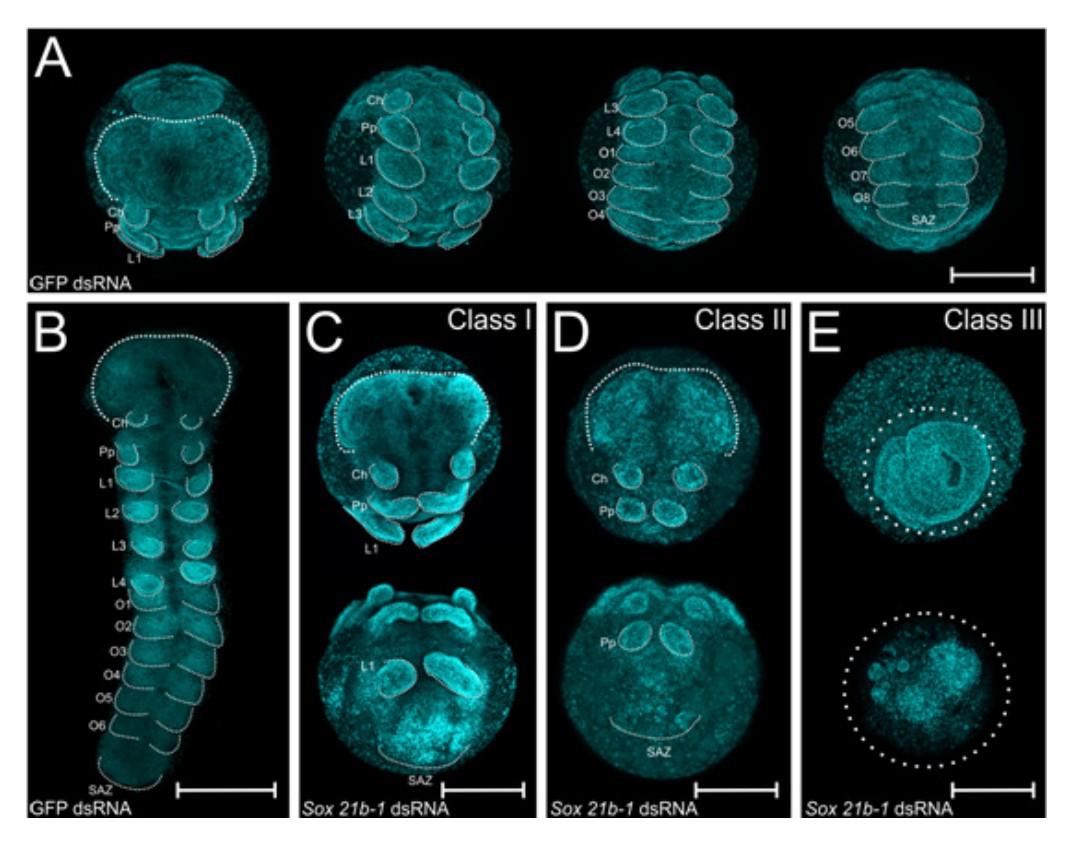

**Figure 2.** Embryo phenotypes after *Sox21b-1* parental RNAi knockdown. Whole mount (**A**) and flat mount (**B**) control embryos at stage 9 stained with DAPI. Stage 9, Class I (**C**), Class II (**D**) and Class III (**E**) phenotypes from *Sox21b-1* knockdown. In the control embryos (**A and B**), the head, cheliceral, pedipalpal, prosomal walking limbs, opisthosomal segments and a posterior SAZ are all clearly visible as indicated. (**C**) Class I phenotype embryos show a morphologically normal head, and pairs of chelicerae, pedipalps and first walking limbs, but a disorganised cluster of cells in the posterior where L2-L4, opisthosomal segments and the SAZ should be. (**D**) Class II phenotype embryos consist of fewer cells, but still form a head, chelicerae, pedipalps and a structure resembling the SAZ in the posterior. (**E**) Class III embryos exhibit the most severe phenotype, where, after the germ disc stage, the embryo fails to form an organised germ band. Ch: Cheliceral segment; L1-L4: Prosomal segments 1 to 4; O1-O6: opisthosomal segments 1 to 6; Pp: Pedipalpal segment; SAZ: Segment Addition Zone. Anterior is to the top, scale bars: 150 μm.

DOI: https://doi.org/10.7554/eLife.37567.006

The following figure supplements are available for figure 2:

**Figure supplement 1.** dsRNA design and phenotypical class frequencies for each fragment and GFP control injections.
DOI: https://doi.org/10.7554/eLife.37567.007

**Figure supplement 2.** Homeotic and mesodermal gene expression at stage 9 in *Sox21b-1* pRNAi embryos.
DOI: https://doi.org/10.7554/eLife.37567.008

**Figure supplement 3.** Snapshots from live imaged videos in control and *Sox21b-1* knockdown embryos.
DOI: https://doi.org/10.7554/eLife.37567.009

**Figure supplement 4.** Cell death and cell proliferation in *Sox21b-1* knockdown embryos.
DOI: https://doi.org/10.7554/eLife.37567.010

prosoma and opisthosomal segments produced from the SAZ were missing (*Figure 2D*). In Class III embryos, the germ band did not form properly from the germ disc (*Figure 2E*) and we therefore looked earlier in development to understand how this severe phenotype arose. We observed that the formation of the primary thickening occurs normally at stage 4 (*Akiyama-Oda and Oda, 2003*; *Pechmann, 2016*; *Pechmann et al., 2017*), but subsequently the cumulus, the group of mesenchymal cells that arise as the primary thickening at the centre of the germ disc, fails to migrate properly to the rim of the germ disc during stage 5 (*Figure 2—figure supplement 3*). Since migration of the cumulus is required for the transition from germ disc to germ band, this observation at least in part explains the subsequent Class III phenotype. Note that in this phenotypic class, the cells migrate

towards the centre of the germ disc, creating a thick aggregation of blastomeres, whereas in embryos that we classified as 'dead', the cells are scattered and stop migrating after the germ disc stage (*Figure 2—figure supplement 3B–C*).

We next examined the effect of *Sox21b-1* depletion on cell death and proliferation at stages 5 and 9 between knockdown and control embryos using antibodies against Caspase-3 and phosphory-lated Histone 3 (PHH3) (*Figure 2—figure supplement 4*). At the germ disc stage, there is no detect-able cell death in control embryos (n = 10), but we observed some small clusters of apoptotic cells in the *Sox21b-1* knockdown embryos (n = 10) (*Figure 2—figure supplement 4A–B*). At stage 9, a few cells expressed Caspase-3 in the posterior-most part of the SAZ (*Figure 2—figure supplement 4C*), but we did not observe cell death in this region of *Sox21b-1* knockdown embryos (*Figure 2—figure supplement 4D*). However, we did detect pronounced cell death in the anterior extraembry-onic layer of the same embryos (n = 10) (*Figure 2—figure supplement 4D*).

Expression of PHH3 at stages 5 and 9 indicated that *Sox21b-1* knockdown embryos show decreased cell proliferation compared to controls (n = 10 for each) (*Figure 2—figure supplement 4E–H*). Interestingly, the cells were also clearly larger in *Sox21b-1* knockdown embryos compared to controls, which may reflect perturbed cell proliferation (*Figure 2—figure supplement 4E–H*). Thus, our functional analysis shows that *Sox21b-1* is required for cell maintenance in several areas of the germ disc and is thus a key player in the transition from radial to axial symmetry. Moreover, *Sox21b-1* is involved in two different segmentation mechanisms in spiders: it has a gap gene like function in the prosoma, as well as a requirement for the formation of the SAZ and subsequent production of opisthosomal segments.

## Effects of *Sox21b-1* knockdown on the germ disc and mesoderm

In *P. tepidariorum*, *decapentaplegic* (*dpp*) and *Ets4* are required for cumulus formation (*Pechmann et al., 2017*; *Akiyama-Oda and Oda, 2006*). To investigate if *Sox21b-1* is involved in the formation of this cell cluster, we assayed the expression of *dpp* and *Ets4* in *Sox21b-1* RNAi knockdown embryos. However, both genes were expressed normally and cumulus formation was unaffected (*Figure 3E,F*).

The rim of the spider germ disc develops into the head structures and is regulated in part by *hh*, while the mesodermal and endodermal layers of the head are specified by the mesendodermal gene *forkhead* (*fkh*) (*Kanayama et al., 2011*; *Akiyama-Oda and Oda, 2010*). To investigate if anterior expression of *Sox21b-1* (*Figure 1*) is involved in the formation of the head rudiment and differentia-tion of the mesodermal and endodermal layers in particular, we assayed the expression of *hh* and *fkh* in class I and II *Sox21b-1* knockdown embryos. *hh* is expressed at the rim of the germ disc in the ectoderm (*Figure 3D*) (*Kanayama et al., 2011*) and remains unaffected by *Sox21b-1* knockdown (*Figure 3H*). *fkh* is also expressed in cells around the rim, as well as in the centre of the germ disc in mesendodermal cells (*Figure 3C*). In *Sox21b-1* knockdown embryos both of these *fkh* expression domains are lost (*Figure 3G*), and it therefore appears that *Sox21b-1* is required for specification of mesendodermal cells in the germ disc of spider embryos. Indeed, in the germ disc at stage 5, when *fkh* expression commences, we observed invaginating cells forming a second layer (*Figure 2—figure supplement 2G*). However, in *Sox21b-1* knockdown embryos we observed a lower number of invagi-nating cells, which exhibit larger nuclei compared to controls (*Figure 2—figure supplement 2H*).

In both spiders and flies, the *twist* (*twi*) gene is involved in mesoderm specification (*Yamazaki et al., 2005*) and we therefore examined the expression of this gene after *Sox21b-1* knockdown to further evaluate if the loss of *fkh* affects the formation of the internal layers. In the wild type, *twi* is expressed in the visceral mesoderm of the limb buds from L1 to L4, in the opisthoso-mal segments O1 to O4 and in an anterior mesodermal patch in the central part of the developing head (*Yamazaki et al., 2005*) (*Figure 2—figure supplement 2E*). While the head expression persists in *Sox21b-1* class I embryos, expression in all the limb and opisthosomal segments is lower or absent (*Figure 2—figure supplement 2F*). In orthogonal projections the anterior-most region of the embryo, three layers of cells can be identified in control embryos (*Figure 2—figure supplement 2I*). However, in *Sox21b-1* knockdown embryos the formation of these layers is perturbed (*Figure 2—figure supplement 2J*). These data suggest that the ectodermal segmentation in the prosomal region occurs even when there is a reduction in the internal layers of the embryo.

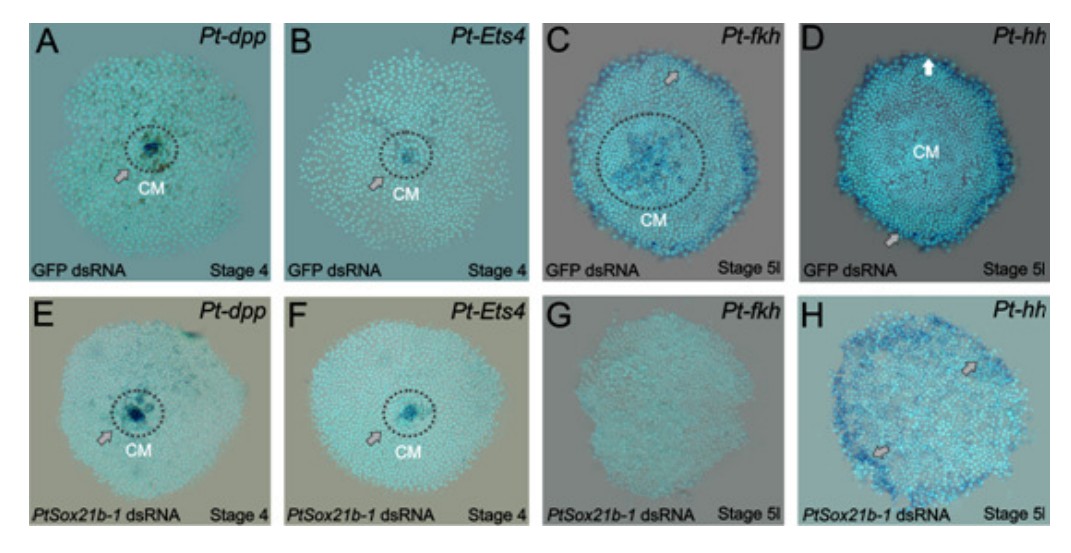

**Figure 3.** Gene expression in control and *Sox21b-1* knockdowns at the germ disc stage. *Pt-dpp* (**A**) and *Pt-Ets4* (**B**) are expressed in the forming cumulus (CM) in the centre of the germ disc at stage 4 (grey arrow and dotted circle). This expression is unaffected by knockdown of *Sox21b-1* (**E** and **F**) (n = 30 for each gene). (**C**) *Pt-fkh* is expressed at the rim and centre of the germ disc at late stage 5 (grey arrow and dotted circle in **C**), but expression is lost in *Sox21b-1* knockdown embryos (n = 30) (**G**). *Pt-hh* expression at the rim of the germ disc (**D**) is normal in *Sox21b-1* knockdown embryos (**H**) (grey arrows).

DOI: https://doi.org/10.7554/eLife.37567.011

## Effects of *Sox21b-1* knockdown on segmentation

In *P. tepidariorum*, formation of the SAZ and production of posterior segments requires the Wnt8 and Delta-Notch signalling pathways (*McGregor et al., 2008b*; *Schönauer et al., 2016*). Interactions between these pathways regulate *hairy* (*h*) and, via *cad,* the expression of pair-rule gene orthologues including *eve* (*McGregor et al., 2008b*; *Schönauer et al., 2016*). To better understand the loss of segments we observe in *Sox21b-1* knockdown embryos we analysed the expression of *Dl, Wnt8, h* and *cad* in these embryos compared to controls.

*Dl* is expressed at stage 7 in the forming SAZ, in the region of the L4 primordia and in the presumptive head (*Oda et al., 2007*) (*Figure 4A*). Subsequently, at stage 9, *Dl* expression is visible in clusters of differentiating neuronal cells and oscillates in the SAZ, an expression pattern associated with the sequential addition of new segments (*Figure 4B*). In *Sox21b-1* knockdown embryos, *Dl* expression is not detected at stage 5 (*Figure 4C*) and is absent in the posterior at stage 9 (*Figure 4D*). However, expression in the anterior neuroectoderm appears normal up to the pedipalpal segment, although neurogenesis is apparently perturbed in the presumptive L1 segment (*Figure 4D*). This suggests that the ectoderm up to the L1 segment differentiates normally, but the development of the SAZ and posterior segment addition controlled by *Dl* is lost upon *Sox21b-1* knockdown.

*Wnt8* is initially expressed at stage 5 in the centre and at the rim of the germ disc (*Figure 4E*). At stage 9, striped expression of *Wnt8* is seen from the head to the posterior segments and in the posterior cells of the SAZ (*Figure 4G*). Knockdown of *Sox21b-1* results in the loss of *Wnt8* expression in late stage 5 embryos (*Figure 4F*). At stage 9, *Wnt8* expression is observed in the cheliceral, pedipalpal and first walking limb segments of *Sox21b-1* knockdown embryos, but no expression is detected in the remaining posterior cells (*Figure 4H*). Consistent with the loss of *Dl* and *Wnt8*, *cad* expression is also lost in stage 5 and stage 9 *Sox21b-1* knockdown embryos (*Figure 4I–L*). These observations indicate that *Sox21b-1* acts upstream of Wnt8 and Delta-Notch signalling to regulate the formation of the SAZ and the subsequent production of posterior segments. In support of this regulatory relationship, we find that *Sox21b-1* expression is still detected in the posterior regions of the truncated embryos produced by RNAi knockdown of either *Dl* or *Wnt8* (*Figure 5*).

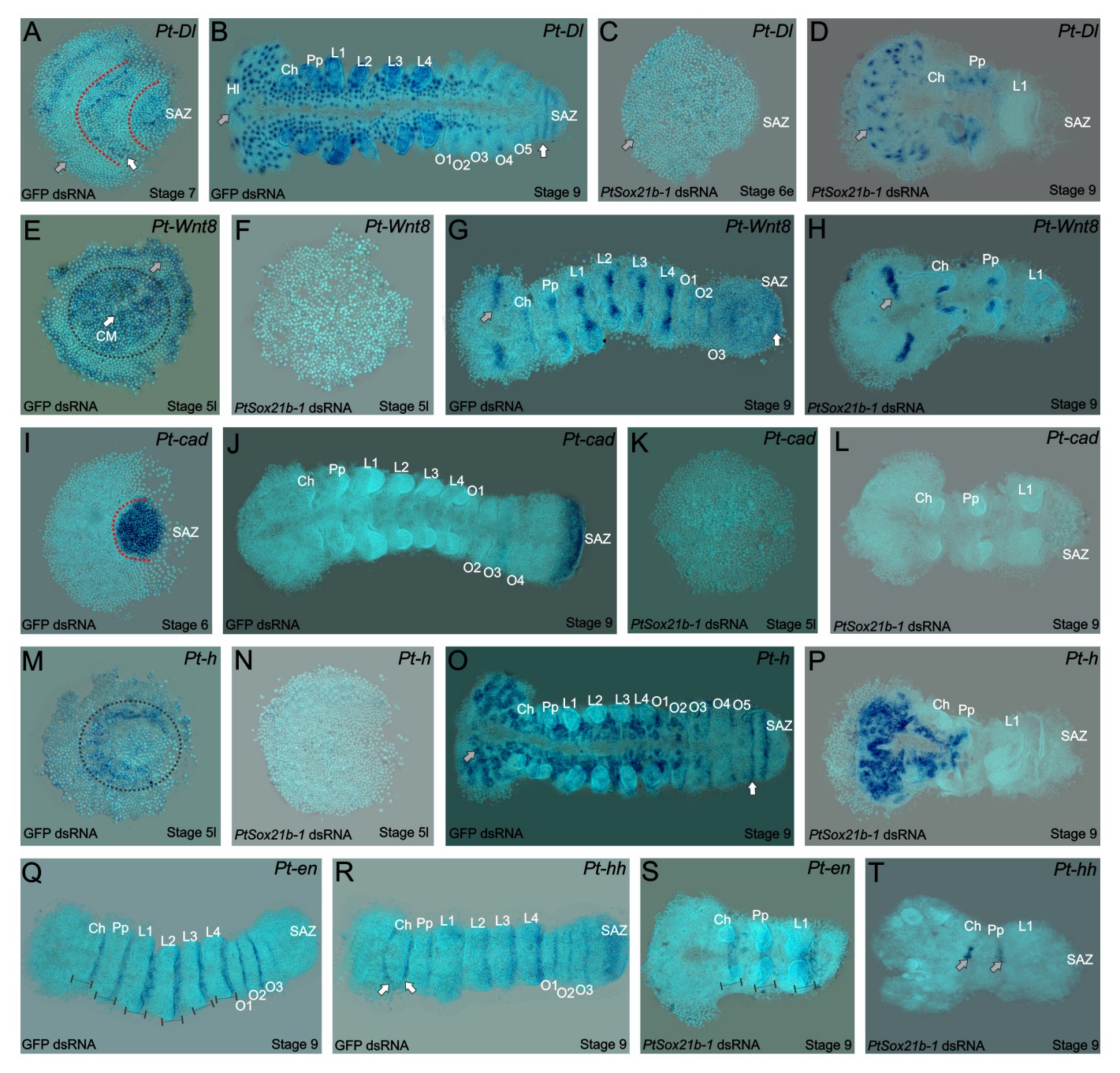

**Figure 4.** Expression of segmentation genes in *Sox21b-1* pRNAi embryos. (**A**) *Pt-Dl* expression at late stage 6/early stage 7 is dynamic in the SAZ and is also observed in the presumptive head region (grey arrow) and prosoma of the embryo (red dotted lines and white arrows). (**B**) At stage 9, *Pt-Dl* expression is seen in the SAZ (white arrow) but is restricted to the clusters of proneural differentiation in the anterior region of the embryo (anterior indicated by a grey arrow). (**C**) In *Sox21b-1* knockdown embryos, *Pt-Dl* expression is not detectable in late stage 5/early stage 6 embryos (anterior indicated by a grey arrow) but can still be observed in the anterior ventral neuroectoderm at stage 9 (**D**) up to the pedipalpal segment (n = 17 and n = 14 for stage 5/6 and 9 respectively). *Pt-Wnt8* expression is observed in the centre and at the rim of the germ disc in stage 5 control embryos (black dotted circle around the centre, grey arrow indicating the rim) but these expression domains are lost in *Sox21b-1* knockdown embryos (n = 11) (**E** and **F**). Control embryos at stage 9 show the expression of *Pt-Wnt8* in the medial region of the head (grey arrow in **G**), and in distal parts of each segment up to the SAZ (white arrow in **G**). In *Sox21b-1* knockdown embryos at the same stage, the brain (grey arrow), chieliceral and pedipalpal expression is still present, but the posterior expression is lost (**H**) (n = 17). *Pt-cad* is expressed in the SAZ at late stage 5/early stage 6 (**I**), which persists to stage 9 in control embryos (**J**). However, *Pt-cad* expression is lost upon *Sox21b-1* knockdown in embryos of both stages (n = 20 for each stage) (**K** and **L**). *Pt-h* expression at stage 5 in control embryos is seen at the rim and in the centre of the germ disc (black dotted circle in **M**), which is lost in *Sox21b-1*
*Figure 4 continued on next page*

*Figure 4 continued*

knockdown embryos (**N**). At stage 9, *Pt-h* expression resembles *Pt-Dl*, both in the control (anterior is indicated by a grey arrow; the SAZ is indicated by a white arrow) and *Sox21b-1* knockdown embryos (**O and P**) (n = 15 for both stages). *Pt-en* expression is present in the posterior of each segment (black lines in **Q**), and in cheliceral, pedipalpal and L1 segments in *Sox21b-1* knockdown embryos at stage 9 (**S**) (n = 10). *Pt-hh* expression in control embryos at stage 9 is seen in the posterior of each segment, in the SAZ and also in a splitting wave between the cheliceral and pedipalpal segments (indicated by the white arrows in **R**). When *Sox21b-1* is knocked-down, *Pt-hh* embryos show expression in the middle-posterior region of the cheliceral and pedipalpal segments (**T**) (n = 8). Ch: Chelicerae; Hl: Head Lobes; L1 to L4: Prosomal leg-bearing segments; O1 to O5: Opisthosomal segments; SAZ: Segment Addition Zone. Anterior is to the left in stage 9 embryos.

DOI: https://doi.org/10.7554/eLife.37567.012

The spider orthologue of *h* is expressed in the presumptive L2-L4 segments and dynamically in the SAZ (*McGregor et al., 2008b*) (*Figure 4M,O*). In late stage 5 *Sox21b-1* knockdown embryos, the expression of *h* is lost throughout the entire germ disc (*Figure 4N*). In addition, in Class I phenotype embryos at stage 9, the expression of *h* is completely absent in the tissue posterior to the pedipalpal segment (*Figure 4P*). Therefore, the loss of *h* expression is consistent with the loss of leg-bearing segments in the anterior gap-like phenotype that results from knockdown of *Sox21b-1* as well as loss of segments produced by the SAZ.

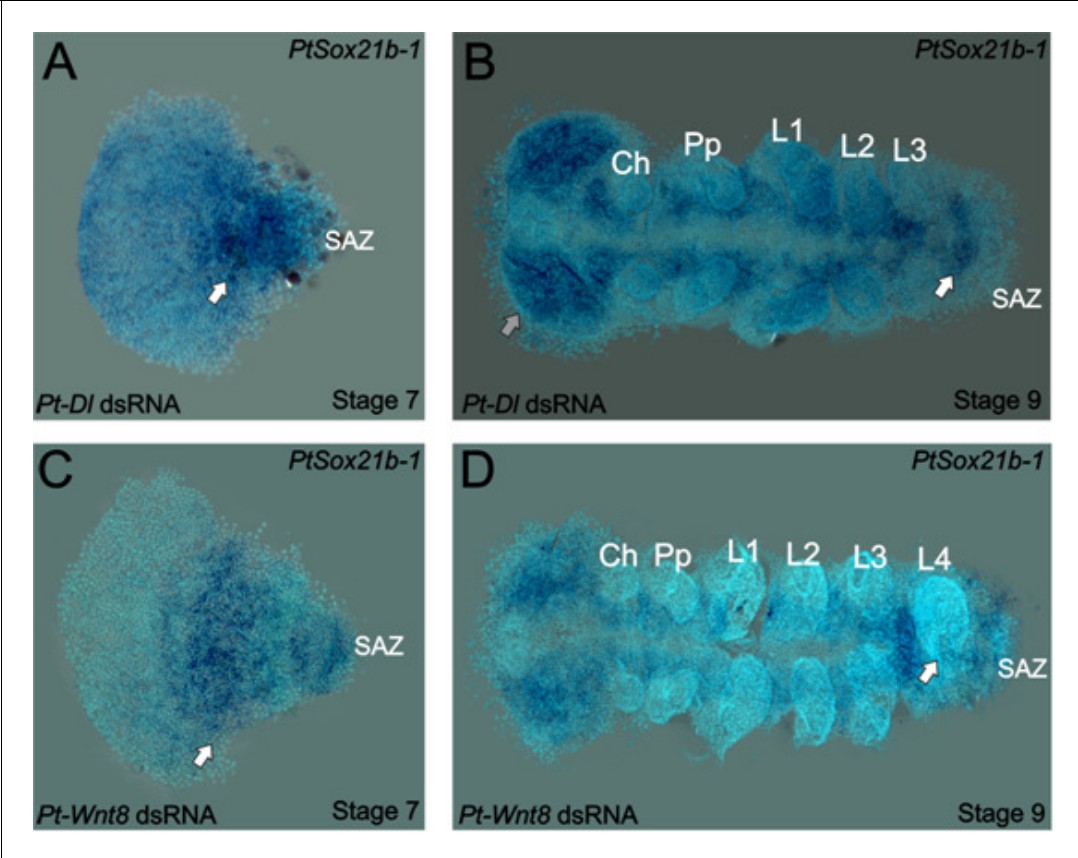

**Figure 5.** Expression of *Sox21b-1* in *Dl* and *Wnt8* pRNAi embryos. Ventral view of stage 7 and 9 knockdown embryos for *Pt-Dl* (**A and B**) and *Pt-Wnt8* (**C and D**). In knockdown embryos for both *Pt-Dl* and *Pt-Wnt8*, *Sox21b-1* is still expressed at stage 7 (**A and C**) in the remaining SAZ cells, and in the forming segments of the presumptive prosoma of the embryo (white arrows). In stage 9 *Pt-Dl* knockdown embryos, *Sox21b-1* remains highly expressed in the ventral nerve cord (**B**). *Pt-Dl* knockdown embryos lack the posterior L4 segment (white arrow), but brain formation appears normal (grey arrow) (**B**). *Pt-Wnt8* embryos show a fusion of the L4 limb buds (white arrow), and *Sox21b-1* is still expressed in the remaining SAZ cells (**D**). Anterior is to the left in all panels.

DOI: https://doi.org/10.7554/eLife.37567.013

To look at the effect of *Sox21b-1* knockdown on segmentation in more detail we examined the expression of *engrailed* (*en*) and *hh*. At stage 9 *en* is expressed segmentally from the cheliceral to the O3 segment in control embryos (*Figure 4Q*). However, in *Sox21b-1* knockdown embryos, expression of *en* was only observed in the cheliceral, pedipalpal and L1 segments, consistent with the loss of all the more posterior segments (*Figure 4S*). *hh* has a similar expression pattern to *en* at stage 9, except it exhibits an anterior splitting wave in the cheliceral segment and is also expressed earlier in opisthosomal segments and in the SAZ (*Figure 4R*). Upon *Sox21b-1* knockdown, *hh* is only detected in shortened stripes in the cheliceral and pedipalpal segments (*Figure 4T*).

Taken together, our analysis of *P. tepidariorum* embryos where *Sox21b-1* is depleted by parental RNAi reveals an important role for this Group B Sox gene in both gap-like segmentation of the prosoma, as well as posterior segment formation from the SAZ. These experiments further emphasise the critical role this class of transcription factors play in arthropod segmentation.

## Discussion

### A SoxB gene is required for two different mechanisms of spider segmentation

The Sox gene family encodes transcription factors that regulate many important processes underlying the embryonic development of metazoans (*Overton et al., 2002*; *Wegner, 1999*; *Sinclair et al., 1990*; *Lefebvre, 2010*). One such gene, *Dichaete,* is expressed in a gap gene pattern and is involved in regulating the canonical segmentation cascade in *D. melanogaster* (*Nambu and Nambu, 1996*; *Russell et al., 1996*). Recently, the analysis of the expression of *Dichaete* in the flour beetle *T. castaneum* strongly suggests a role in short germ segmentation (*Clark and Peel, 2018*), further supported by knockdown of the *Dichaete* orthologue in *Bombyx mori,* which resulted in the loss of posterior segmentation (*Nakao, 2018*).

Here we show that, while *Dichaete* is not involved in spider segmentation (*Paese et al., 2017*), the closely related SoxB gene, *Sox21b-1*, regulates formation of both prosomal and opisthosomal segments. In the prosoma *Sox21b-1* has a gap gene like role and is required for the specification of L1-L4 segments (*Figure 6*), resembling the roles of *hb* and *Dll* in prosomal segmentation in this spider (*Pechmann et al., 2011*; *Schwager et al., 2009*) and, at least superficially, gap gene function in *D. melanogaster.*

In *D. melanogaster* the gap genes regulate pair rule gene expression, and while our results indicate that *Sox21b-1* is required for the expression of *h* and the generation of leg-bearing prosomal segments (*Figure 4E*, *Figure 6*), in contrast to insects, in spiders this does not involve the orthologues of *eve* and *runt* because they are not expressed in the developing prosomal segments (*Schönauer et al., 2016*; *Damen et al., 2000*).

In the posterior, *Sox21b-1* knockdown perturbs SAZ formation and consequently results in truncated embryos missing all opisthosomal segments. Therefore, *Sox21b-1* regulates development of the SAZ, and our observations indicate this is at least in part through roles in organising the germ layers and specification of mesendodermal cells during stages 5 and 6. This is supported by the loss of *fkh* expression upon *Sox21b-1* knockdown, which is required for mesoderm and endoderm formation in both spiders and insects (*Kanayama et al., 2011*; *Feitosa et al., 2017*; *Lan et al., 2018*). Moreover, the subsequent dynamic expression of *Sox21b-1* in the SAZ after stage 6 is suggestive of a role in segment addition.

Our work on *Sox21b-1* provides an important new insight into the gene regulatory network (GRN) underlying the formation of the SAZ and the sequential addition of segments from this tissue. We show that *Sox21b-1* acts upstream of Wnt8 and Delta-Notch signalling in this GRN and is necessary for the activation of these important signalling pathways during posterior development (*Figure 6*). Note that while it is possible that Sox21b-1 could regulate *Delta* and *Wnt8* and other segmentation genes directly or via intermediate factors, it remains possible that the loss of expression of these genes upon *Sox21b-1* RNAi could be an indirect consequence of the loss of cells or incorrect cell specification when *Sox21b-1* expression is knocked down.

Further work is needed to determine if Group B Sox genes, such as *Dichaete* and *Sox21b-1*, play a similar role in posterior segmentation in other arthropods. This could provide important new insights into the evolution of the regulation of segmentation in arthropods since a Wnt-Delta-Notch-

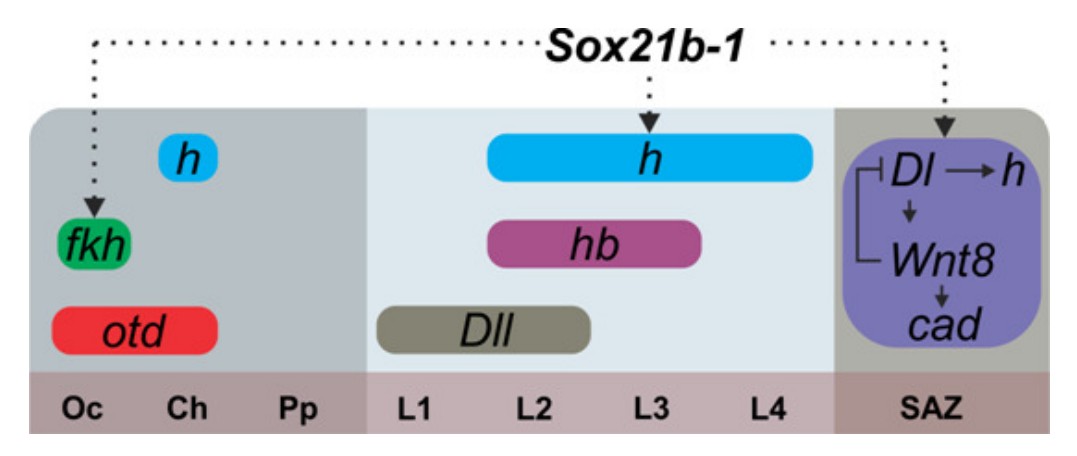

**Figure 6.** Summary of the regulation of spider segmentation. The interaction of *Sox21b-1* is presented in relation to genes involved in spider embryogenesis. We found that *fkh* expression requires *Sox21b-1* in the most anterior part of the head . In the prosoma *Sox21b-1* also has a gap gene like function and is required for the expression of *hairy*, while *Distal-less* (*Pechmann et al., 2011*), *hunchback* and *orthodenticle* (*Pechmann et al., 2009*) also act in a gap gene like manner during prosomal segmentation. The molecular control of segmentation in the SAZ involves a feedback loop between *Dl* and *Wnt8*, which acts upstream of *cad* and also controls the dynamic expression of *hairy* (*Schönauer et al., 2016*; *McGregor et al., 2008b*). We can infer from our results that *Sox21b-1* acts upstream of these genes in the SAZ. Sox21b-1 could directly regulate these genes or the observed loss of expression could be indirect through incorrect specification of germ disc cells. Oc: Ocular segment; Ch: Cheliceral segment; Pp: Pedipalpal segment; L1-L4: leg bearing segments 1 to 4; SAZ: Segment Addition Zone.
DOI: https://doi.org/10.7554/eLife.37567.014

Cad regulatory cassette was probably used ancestrally in arthropods to regulate posterior development (*McGregor et al., 2009*; *Janssen et al., 2004*; *Brena and Akam, 2012*; *McGregor et al., 2008b*; *Schönauer et al., 2016*; *Pueyo et al., 2008*). Interestingly, SoxB genes also cooperate with Wnt and Delta-Notch signalling in various aspects of vertebrate development including the patterning of neural progenitors and maintenance of the stem state in the neuroepithelium (*Wegner, 1999*; *Holmberg et al., 2008*; *Kormish et al., 2010*; *Koch et al., 2017*).

## *Sox21b-1* exhibits highly pleiotropic effects during early spider embryogenesis

Our study shows that *Sox21b-1* is not only involved in segmentation but is also maternally supplied and regulates cell division in the early germ disc, as well as the transition from radial to axial symmetry during germ band formation. Further experiments with *Sox21b-1* are required to fully elucidate the mechanisms by which it affects these early functions. Furthermore, while spider head development is less affected than trunk segmentation by knockdown of *Sox21b-1*, it is clear from our experiments that *Sox21b-1* regulates cell fate in this region. Interestingly, Sox2 is involved with the neuro-mesodermal fate choice in mice and *Dichaete* has a role in embryonic brain development in *D. melanogaster* (*Koch et al., 2017*; *Soriano and Russell, 1998*): consequently, SoxB genes may play an ancestral role in the patterning of the head ectoderm and mesoderm in metazoans (*Koch et al., 2017*; *Soriano and Russell, 1998*).

## The evolution of *Sox21b-1*

The evolution and diversification of Group B Sox genes in insects is not fully resolved due to difficulties in clearly assigning orthologues based on the highly conserved HMG domain sequence (*Russell et al., 1996*; *Wegner, 1999*; *Zhong et al., 2011*). However, despite these ambiguities it is clear that the *Dichaete* and *Sox21b* class genes in all arthropods examined to date are closely related and likely arose from a duplication in the common ancestor of this phylum (see *Zhong et al., 2011* for discussion). Note that in all insects characterised to date *Dichaete*, *Sox21a* and *Sox21b* are clustered in the genome (*McKimmie et al., 2005*), however, while *Dichaete* and *Sox21a* are also clustered in *P. tepidariorum*, the *Sox21b* paralogs are dispersed in the genome of this spider

(*Paese et al., 2017*). We believe it is highly significant that two very closely related SoxB genes are involved in segmentation in both the spider *P. tepidariorum* and in insects, pointing to an ancient role for this subfamily of Sox genes in invertebrates. Given the close similarity between the HMG domains of Sox21b and Dichaete, it is possible that in some lineages the Dichaete orthologue assumed the segmentation role, whereas in others it was Sox21b. In spiders, *Wnt8* is involved in posterior development while in other arthropods this role is played by *Wnt1/wg* (*McGregor et al., 2008b*), and therefore the evolution of *Sox21b-1* may have led to the co-option of different genes and subsequent developmental systems drift of the regulation of posterior development.

The spider contains an additional related SoxB gene, *Sox21b-2*, that possibly arose as part of the whole genome duplication event in the ancestor of arachnopulmonates over 400 million years ago (*Schwager et al., 2017*). It will be interesting to investigate if SoxB genes are involved in segmentation in other spiders and arachnids, including those that did not undergo a genome duplication. Finally, Blast searches of the Tardigrade *Hypsibius dujardini* genome reveal a single *Dichaete/Sox21b* class gene, and it will be of some interest to characterise the expression and/or function of this gene in this sister group to the arthropods.

## Materials and methods

**Key resources table**

| Reagent type (species) or resource | Designation | Source or reference | Identifiers | Additional information |
|---|---|---|---|---|
| Gene (*Parasteatoda tepidariorum*) | caudal | NA | AB096075 | |
| Gene (*P. tepidariorum*) | deformed-a | NA | AB433904 | |
| Gene (*P. tepidariorum*) | decapentaplegic | NA | AB096072 | |
| Gene (*P. tepidariorum*) | delta | NA | AB287420 | |
| Gene (*P. tepidariorum*) | engrailed | NA | AB125741 | |
| Gene (*P. tepidariorum*) | ets4 | NA | XP_015923392 | |
| Gene (*P. tepidariorum*) | forkhead | NA | AB096073 | |
| Gene (*P. tepidariorum*) | hairy | NA | AB125743 | |
| Gene (*P. tepidariorum*) | hedgehog | NA | AB125742 | |
| Gene (*P. tepidariorum*) | labial | NA | AB433903 | |
| Gene (*P. tepidariorum*) | sox21b-1 | NA | XP_015916301 | |
| Gene (*P. tepidariorum*) | twist | NA | AB167807 | |
| Gene (*P. tepidariorum*) | Wnt8 | NA | ACH88002 | |
| Antibody | Donkey anti-mouse IgG Alexa Fluor 555 | Invitrogen | A-31570 | (1:200) |
| Antibody | Goat anti-rabbit Alexa Fluor 647 | Invitrogen | A-21244 | (1:200) |
| Antibody | Mouse anti-α-Tubulin | Sigma | DM1a | (1:50) |
| Antibody | Rabbit Anti-phospho-Histone H3 (Ser10) | Merck Millipore | 06–570 | (1:200) |

*Continued on next page*

*Continued*

| Reagent type (species) or resource | Designation | Source or reference | Identifiers | Additional information |
|---|---|---|---|---|
| Antibody | Rabbit α cleaved caspase 3 | Cell Signaling | 9661 | (1:200) |
| Chemical compound, drug | 4′,6-Diamidino-2-phenylindole dihydrochloride | Sigma-Aldrich | D8417 | |
| Chemical compound, drug | Halocarbon Oil 700 | Sigma-Aldrich | H8898 | |
| Chemical compound, drug | Poly-L-lysine | Sigma-Aldrich | P9155 | |
| Chemical compound, drug | TWEEN 20 | Sigma-Aldrich | P9416 | |
| Commercial assay or kit | TOPO-TA Cloning Kit | Invitrogen | K457502 | |
| Software, algorithm | Corel Graphics Suite | | RRID:SCR_013674 | |
| Software, algorithm | Code used for genomics data analysis - PROTGAMMALG | | | https://github.com/stamatak/standard-RAxML/blob/master/usefulScripts/ProteinModelSelection.pl |
| Software, algorithm | ImageJ | | RRID:SCR_003070 | http://imagej.nih.gov/ij/ |
| Software, algorithm | Helicon Focus | | RRID:SCR_014462 | |

## Spider culture

*P. tepidariorum* were cultured at 25°C at Oxford Brookes University. The spiders were fed with *D. melanogaster* with vestigial wings and subsequently small crickets (*Gryllus bimaculatus*). Cocoons from mated females were removed and a small number of embryos were immersed in halocarbon oil for staging (*Mittmann and Wolff, 2012*).

## Phylogenetic analysis of *P. tepidariorum* Sox genes

To identify the phylogenetic relationship of *P. tepidariorum* Sox genes the HMG domains of *Anopheles gambiae*, *Mus musculus*, *D. melanogaster*, *P. tepidariorum* and *S. mimosarum* Sox genes were aligned with ClustalW (*Figure 1—source data 1*) (*Paese et al., 2017*; *Larkin et al., 2007*). Phylogenetic analysis was performed in RAxML, with support levels estimated implementing the rapid bootstrap algorithm (1000 replicates) (*Stamatakis et al., 2008*), under the PROTGAMMALG model of amino acid substitution, which was identified as best fitting model using the ProteinModelSelection.pl Perl script from the Exelixis Lab (https://github.com/stamatak/standard-RAxML/blob/master/usefulScripts/ProteinModelSelection.pl).

## Fixation and gene expression analysis

Embryos from pRNAi injected females were immersed under halocarbon oil to assess their stage. In the case of stage 9 and 10 embryos that showed class II and III phenotypes, it was difficult to assess the stage by light microscopy, consequently these embryos were also staged according to their age and the presence of walking limbs as well as the stage of sibling embryos from the same cocoons that did not show any phenotype. Embryos with phenotypes showing a failure to develop after the L1 limb were considered dead after that stage. For the difference in development seen in embryos from the same cocoon, we were careful to select embryos at the same stage in both control and RNAi treated embryos for subsequent gene expression analysis although the development of RNAi embryos was occasionally slower so they appeared slightly younger. At least 10 embryos from each

different phenotypical class were used for every in situ hybridization experiment (*Supplementary file 2*) and in situs were carried out on embryos from different cocoons and different mothers. Embryos ranging from the 1 cell stage to stage 13 were dechorionated and fixed as described previously (*Akiyama-Oda and Oda, 2016*), with a longer fixation time of 1 hr to facilitate yolk removal for flat-mounting. For immunohistochemistry, methanol steps were omitted. Ovaries from adult females were dissected in 1x PBS and fixed in 4% formaldehyde for 30 min. Probe synthesis and RNA in situ hybridisation were carried out as described previously with minor modifications (*Akiyama-Oda and Oda, 2003*), omitting the Proteinase K treatment and post-fixation steps. Poly-L-lysine (Sigma-Aldrich) coated coverslips were used for flat-mounting embryos. Nuclei were stained by incubating embryos in 1 µg/ml 4–6-diamidino-2-phenylindol (DAPI) in PBS with 0.1% Tween-20 for 15 min.

## Imaging, live imaging and image analysis

For imaging of flat-mounted embryos after in situ hybridisation, an AxioZoom V16 stereomicroscope (Zeiss) equipped with an Axiocam 506-Mono and a colour digital camera were used. Immunostained embryos were imaged with Zeiss LSM 800 or 880 with Airyscan confocal microscopes. For live imaging, embryos were aligned on heptane glue coated coverslips and submersed in a thin layer of halocarbon oil. Bright-field live imaging was performed using an AxioZoom V16 stereomicroscope, while fluorescence live imaging was performed with confocal microscopes. Image stacks were processed in Fiji (*Schindelin et al., 2012*) and Helicon Focus (HeliconSoft). Image brightness and intensity was adjusted in Corel PhotoPaint X5 (CorelDraw) and Fiji.

## Gene isolation from cDNA

Fragment of genes were amplified using PCR and cloned into pCR4-TOPO (Invitrogen, Life Technologies). Oligonucleotide sequences are listed in *Supplementary file 3*.

## Immunohistochemistry

Immunostaining was carried out as described previously (*Schwager et al., 2015*) with minor modifications: antibodies were not pre-absorbed prior to incubation and the concentration of Triton was increased to 0.1%. The following primary antibodies were used: mouse anti-α-Tubulin DM1a (Sigma) (1:50), rabbit α cleaved caspase 3 (Cell Signaling - 9661) (1:200) and rabbit Anti-phospho-Histone H3 (Ser10) (Merck Millipore - 06–570). For detection the following secondary antibodies were used: donkey anti-mouse IgG Alexa Fluor 555 (Invitrogen) and goat anti-rabbit Alexa Fluor 647 (Invitrogen). The counterstaining was carried out by incubation in 1 µg/ml 4–6-diamidino-2-phenylindol (DAPI) in PBS + Triton 0,1% for 20 min.

## dsRNA synthesis and parental RNA interference

Double stranded RNA (dsRNA) for parental RNA interference was synthesized according to (*Schönauer et al., 2016*), dissolved in deionized water and injected following the standard protocol (*Akiyama-Oda and Oda, 2006*). Two non-overlapping fragments of *P. tepidariorum Sox21b-1* were isolated from the 1134 bp coding sequence of the gene: fragment 1 spanning 549 bp and fragment 2 covering 550 bp. Double stranded RNA for *P. tepidariorum Dl* (853 bp), *Wnt8* (714 bp) and the coding sequence of GFP (720 bp) as used previously (*Akiyama-Oda and Oda, 2006*), were transcribed using the same method. Synthesis of double stranded RNA was performed using the MegaScript T7 transcription kit (Invitrogen). After purification the dsRNA transcripts were annealed in a water bath starting at 95°C and slowly cooled down to room temperature. dsRNA was injected at 2.0 µg/µl in to the opisthosoma of adult females every two days, with a total of five injections (n = 7 for each dsRNA; n = 2 for GFP controls). The injected spiders were mated after the second injection and embryos from injected spiders were fixed for gene expression and phenotypic analyses at three different time points: stage 4 (cumulus formation), stage 5 late (germ disc with migrating cumulus) and stage 9 (head and limbs bud formation).

## Acknowledgements

This research was funded by a CNPq scholarship to CLBP (234586/2014-1), a grant from The Leverhulme Trust (RPG-2016-234) to APM and AS, and in part by a BBSRC grant (BB/N007069/1) to SR.

## Additional information

### Funding

| Funder | Grant reference number | Author |
|---|---|---|
| Conselho Nacional de Desenvolvimento Científico e Tecnológico | 234586/2014-1 | Christian Louis Bonatto Paese |
| Leverhulme Trust | RPG-2016-234 | Alistair P McGregor<br>Anna Schoenauer |
| Biotechnology and Biological Sciences Research Council | BB/N007069/1 | Steven Russell |

The funders had no role in study design, data collection and interpretation, or the decision to submit the work for publication.

### Author contributions

Christian Louis Bonatto Paese, Conceptualization, Formal analysis, Funding acquisition, Validation, Investigation, Visualization, Methodology, Writing—original draft, Writing—review and editing; Anna Schoenauer, Conceptualization, Formal analysis, Investigation, Methodology, Writing—original draft, Writing—review and editing; Daniel J Leite, Formal analysis, Investigation, Methodology, Writing—original draft, Writing—review and editing; Steven Russell, Conceptualization, Supervision, Writing—original draft, Project administration, Writing—review and editing; Alistair P McGregor, Conceptualization, Supervision, Funding acquisition, Writing—original draft, Project administration, Writing—review and editing

### Author ORCIDs

Christian Louis Bonatto Paese (iD) https://orcid.org/0000-0001-5992-5209
Steven Russell (iD) http://orcid.org/0000-0003-0546-3031
Alistair P McGregor (iD) http://orcid.org/0000-0002-2908-2420

### Decision letter and Author response

Decision letter https://doi.org/10.7554/eLife.37567.020
Author response https://doi.org/10.7554/eLife.37567.021

## Additional files

### Supplementary files

• Supplementary file 1. Phenotypic frequencies for each fragment (1 and 2) and GFP (control) dsRNA – 30 embryos per cocoon for each spider were pooled and the characteristics were divided into three phenotypical classes.
DOI: https://doi.org/10.7554/eLife.37567.015

• Supplementary file 2. Sample sizes for the *PtSox21b-1* phenotype analysis with in situ hybridization, and phenotypic classes used in each experiment.
DOI: https://doi.org/10.7554/eLife.37567.016

• Supplementary file 3. List of primers used in this paper.
DOI: https://doi.org/10.7554/eLife.37567.017

• Transparent reporting form
DOI: https://doi.org/10.7554/eLife.37567.018

## Data availability

All data generated or analysed during this study are included in the manuscript and supporting files.

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
