## [Decision Letter]

Thank you for submitting your article "A SoxB gene acts as an anterior gap gene and regulates posterior segment addition in a spider" for consideration by *eLife*. Your article has been reviewed by Patricia Wittkopp as the Senior Editor, a Reviewing Editor, and three reviewers. The following individuals involved in review of your submission have agreed to reveal their identity: Nikola-Michael Prpic (Reviewer #1); Natascha Turetzek (Reviewer #2).

The reviewers have discussed the reviews with one another and the Reviewing Editor has drafted this decision to help you prepare a revised submission.

Summary:

The authors have studied the role of a SoxB related gene, *Sox21b-1*, during the embryonic development of the spider *Parasteatoda tepidariorum*. In insects where the role of SoxB family genes in segmentation was examined, it was reported that a particular member of this gene family, called *Dichaete*, was involved. The present paper shows that although *Dichaete* is present and conserved in spiders, it is not involved in segmentation; instead, another SoxB paralog called *Sox21b-1* is involved in spider segmentation. The authors show that the gene is maternally provided and is then expressed in an anterior and a posterior domain in the forming embryo. They have also tested the function of the gene by RNAi and find that the RNAi knockdown animals show a striking phenotype where the entire posterior portion of the germ band is missing. Interestingly, a milder phenotype shows a "gap-gene-like" phenotype. The authors therefore go on to investigate the influence of *Sox21b-1* RNAi on known gap-like factors and posterior dynamic factors. They convincingly show that *Sox21b-1* is required for the expression of the known gap-like factors but is also required for the expression of the posterior dynamic patterning system that operates in the segment addition zone. Thus, with this work the authors provide the first factor that integrates the central gap system with the posterior dynamic system of spider segmentation. In fact, Paese et al. have identified a gene that is required for the development of the entire opisthosoma, and acts upstream of the previously identified system of caudal, Delta-Notch, and Wnt8 signalling. Moreover, the authors provide data for a mechanism in which *Sox21b-1* orchestrates posterior segment addition by organising the germ layers and the specification of mesendodermal cells. Finally, this study provides insight about how certain gene regulatory interactions can change during evolution following gene/genome duplication events.

Essential revisions:

1) The authors should interpret the missing expression of some likely target genes after *Sox21b-1* RNAi more carefully. The expression might be reduced/lost because the input of *Sox21b-1* is missing. However, the expression might also be lost simply because the cells/tissue/organ where the gene is usually expressed are reduced/lost (especially because some of the RNAi phenotypes are quite severe because of the loss of *Sox21b-1* maternal contribution). By using whole-mount in situ hybridisation as the only indicator of target gene expression, it may be difficult in some cases to distinguish why the expression is reduced/lost in the RNAi animals and this may lead to erroneous statements about gene regulatory interactions between the studied genes. Remarkably, all of the studied (likely) target genes presumably respond "negatively" to *Sox21b-1* RNAi (i.e. their expression is reduced or lost), which increases the likelihood that not all of these "negative responses" are due to true genetic interactions. To clarify this, a marker could be used that co-expresses with one (or all) of these presumed targets, but that is not regulated by *Sox21b-1* to show that the cells are there and their identity properly specified. This would strengthen the authors’ claim that *Sox21b-1* is upstream and solidify their model in Figure 6. Alternatively, clonal RNAi in embryos could be performed, that would provide "control" cells directly next to injected RNAi cells. As these experiments would, however, require extensive additional work, the authors are therefore asked to carefully rephrase those portions of the text that deal with this issue of gene regulation and expressly state that the reduction/loss of gene expression after RNAi may have other causes than gene interaction and therefore not all of the genes shown in the manuscript to respond negatively upon *Sox21b-1* RNAi may be true regulatory targets.

2) The authors provide a very detailed staging of all the presented embryos, however the decision exactly which developmental stage is present is not always clear from the pictures especially in severely affected RNAi embryos. Please provide the criteria for staging of the RNAi embryos in the Materials and methods section. Was the staging based on certain landmarks or developmental processes of the normal developmental table? If yes, how was the decision made when these landmarks were lost after gene knockdown? Or were the RNAi embryos staged by developmental time?

---

## [Author Response]

Essential revisions:1) The authors should interpret the missing expression of some likely target genes after Sox21b-1 RNAi more carefully. The expression might be reduced/lost because the input of Sox21b-1 is missing. However, the expression might also be lost simply because the cells/tissue/organ where the gene is usually expressed are reduced/lost (especially because some of the RNAi phenotypes are quite severe because of the loss of Sox21b-1 maternal contribution). By using whole-mount in situ hybridisation as the only indicator of target gene expression, it may be difficult in some cases to distinguish why the expression is reduced/lost in the RNAi animals and this may lead to erroneous statements about gene regulatory interactions between the studied genes. Remarkably, all of the studied (likely) target genes presumably respond "negatively" to Sox21b-1 RNAi (i.e. their expression is reduced or lost), which increases the likelihood that not all of these "negative responses" are due to true genetic interactions. To clarify this, a marker could be used that co-expresses with one (or all) of these presumed targets, but that is not regulated by Sox21b-1 to show that the cells are there and their identity properly specified. This would strengthen the authors´ claim that Sox21b-1 is upstream and solidify their model in Figure 6. Alternatively, clonal RNAi in embryos could be performed, that would provide "control" cells directly next to injected RNAi cells. As these experiments would, however, require extensive additional work, the authors are therefore asked to carefully rephrase those portions of the text that deal with this issue of gene regulation and expressly state that the reduction/loss of gene expression after RNAi may have other causes than gene interaction and therefore not all of the genes shown in the manuscript to respond negatively upon Sox21b-1 RNAi may be true regulatory targets.

The reviewers raise an important point. While our data shows that Sox21b-1 acts upstream of the segmentation genes we have tested, we did not mean to suggest that these genes are directly regulated by this transcription factor. Indeed, this regulation could well be several steps upstream of these genes and Sox21b-1 regulation could even be indirect by specifying the cells that will express these genes. Unfortunately, we do not know of a suitable marker that is expressed in the specific germ disc cells at the relevant stages that would be able to help with this point as suggested by the reviewers. With regard to embryonic RNAi injections, we believe that clonal analysis of *Sox21b-1* knockdown may be difficult to interpret and, as the reviewers recognise, would involve a significant amount of new experimental effort that, in the end, may not be conclusive. However, as mentioned above, we do accept the reviewers point that effects we see may not be direct and, therefore, as requested, we have rephrased the relevant text to clarify that not all of the genes that respond negatively to the *Sox21b-1* RNAi may be regulatory targets and that indeed the observed effects could be indirect through Sox21b-1 regulation of other intermediate genes or through the incorrect specification of germ disc cells. For example, in the Discussion section, we have added: “Note that while it is possible that Sox21b-1 could regulate *Delta* and *Wnt8* and other segmentation genes directly or via intermediate factors, it remains possible that the loss of expression of these genes upon *Sox21b-1* RNAi could be an indirect consequence of the loss of cells or incorrect cell specification when *Sox21b-1* expression is knocked-down.”

We have also tried to make this clearer in Figure 6 and the legend of this figure.

2) The authors provide a very detailed staging of all the presented embryos, however the decision exactly which developmental stage is present is not always clear from the pictures especially in severely affected RNAi embryos. Please provide the criteria for staging of the RNAi embryos in the Materials and methods section. Was the staging based on certain landmarks or developmental processes of the normal developmental table? If yes, how was the decision made when these landmarks were lost after gene knockdown? Or were the RNAi embryos staged by developmental time?

This is a very good point raised by the reviewers. Therefore, we have added a more detailed explanation of the criteria used for the fixation and staging of the RNAi embryos to the Materials and methods section. In summary, even though the embryos were immersed in oil for visualization, it was sometimes difficult to stage embryos due to their severe phenotypes (especially class III). Consequently, we relied on developmental staging by hours, and the morphology of sibling embryos from the same cocoon that did not show the phenotype (i.e. from cocoons where the effect was not fully penetrant) and at least in the case of older embryos, by the appearance of the limbs.